# Numerical Investigation of Uplift Failure Mode and Capacity Estimation for Deep Helical Anchors in Sand

Chi Yuan [1], Dongxue Hao [2,3,*], Rong Chen [2,3,*] and Ning Zhang [1]

1  College of Architecture and Civil Engineering, Beijing University of Technology, Beijing 110124, China; yuanc@emails.bjut.edu.cn (C.Y.)
2  Key Lab of Electric Power Infrastructure Safety Assessment and Disaster Prevention of Jilin Province, Northeast Electric Power University, Jilin 132012, China
3  School of Civil Engineering and Architecture, Northeast Electric Power University, Jilin 132012, China
*  Correspondence: 20102291@neepu.edu.cn (D.H.); 20112384@neepu.edu.cn (R.C.); Tel.: +86-432-6480-6481 (D.H.)

**Abstract:** The uplift capacity of helical anchors is generally taken as the control condition for design in different applications, including transmission tower foundations and offshore structures. However, it is difficult to identify the failure surface for a deep helical anchor, which may result in an incorrect assessment of uplift capability. This research proposes a new unified method to estimate the uplift capacity of deep single-helix and multi-helix anchors based on the investigation of failure mechanisms. The deep failure mode was identified by FEM analysis using a modified Mohr–Coulomb model considering the strain softening of sand, along with the coupled Eulerian–Lagrangian technique. Thereby, a simplified rupture surface is proposed, and the equations estimating the uplift capacity are presented by the limit equilibrium method. Two important factors—the lateral earth pressure coefficient and the average internal friction angle included in the equations—are discussed and determined. The comparisons with centrifugal tests verify the reasonability of the proposed method.

**Keywords:** helical anchor; CEL analysis; deep failure mode; uplift capacity

## 1. Introduction

A helical anchor, consisting of one or more helical plates welded to a steel shaft, is a deep foundation system used to support or resist any load or application. Due to their rapid installation and immediate service, relatively large bearing capacity, and lack of environmental damage, they have been extensively employed as a foundation system for structures such as transmission towers, offshore platforms, and wind turbines [1,2]. In recent years, this type of foundation has been suggested as a potential alternative to driven piles in offshore renewable energy structures.

There are three failure modes for single-helix anchors as the embedment depth ratio ($H/D$) increases, where $D$ is the helix diameter. Figure 1 shows the shallow failure mode [3–12], deep failure mode [11,13–16], and transition failure mode [4,13] of circular plate anchors or single-helix anchors, which have been observed in most investigations. The rupture surface extends continuously to the ground at a shallow plate depth, defined as general shear failure or the shallow failure mode. With the increase in plate depth, the transition failure mode is observed, often referred to as the shallow failure mode [4,13]. The shallow rupture surface has been assumed to be a cylinder [3], an inverted cone [7,8,13], or a log-spiral surface [5,10,17]. Only a few small-scale model tests showed a closed bulb (or balloon-shaped) rupture surface [13–16,18]. The rupture surface is limited under the ground at a deep depth, defined as local shear failure or the deep failure mode. Recently, a centrifuge test with $H/D = 7$ observed the deep rupture surface in medium dense sand [19]. Although it provided intuitive observation, there could be a discrepancy between the results of the

half-anchor model and the full-anchor model due to the difference in sand–strongbox interaction from the internal friction of sand.

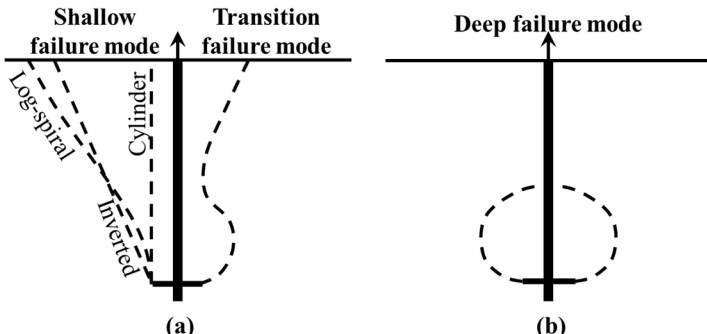

**Figure 1.** The failure mode of a single-helix or plate anchor. (**a**) Shallow and transition failure mode. (**b**) Deep failure mode.

It is generally believed that there are two failure modes for multi-helix anchors controlled by the helical plate space ratio ($S/D$)—that is, the cylindrical shear [4,20–22] and the individual bearing failure mode [23,24], as shown in Figure 2. The cylindrical shear failure mode means that the cylindrical rupture surface extends to the uppermost plate and then follows the single-helix anchor failure mode. The individual bearing failure mode means that each helical plate behaves independently of the others. The transition helix spacing between the two methods is usually regarded as $3D$ in some engineering manuals [23,25,26].

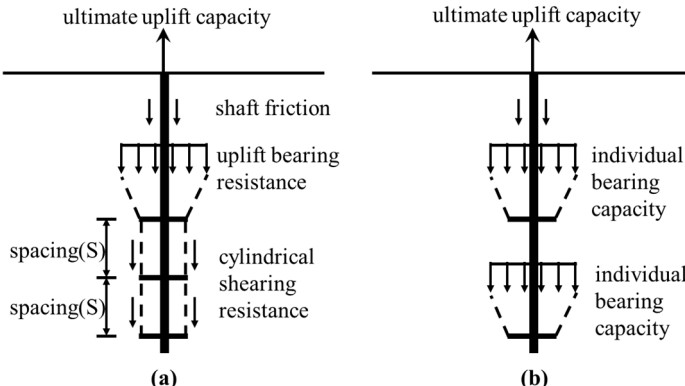

**Figure 2.** The failure mode of a multi-helix or plate anchor. (**a**) Cylindrical shear mode. (**b**) Individual bearing mode.

There are two issues that need further discussion for deep helical anchors in sand:

(1) The direct observations of rupture surfaces are limited, especially for the multi-helix anchors. Some observations on the rupture surface of single-helix or circular plate anchors in sand have been reported, but they mostly came from 1 g small-scale model tests. This may produce differences between the observed rupture surface and the actual state due to the low stress level, especially for deep anchors. Therefore, it is necessary to further study the failure mode of deep helical anchors, which is essential for the estimation of the uplift capacity.

(2) The estimation of uplift capacity for multi-helix anchors with transition helix spacing based on the two recognized failure modes (Figure 2) is inconsistent. That is, when the helix spacing is transition spacing, the uplift capacity calculated by the individual bearing method is higher than that calculated by the cylindrical shear method.

Numerical simulation is an effective option to analyze the failure mechanisms and bearing capacity of foundations. The finite element method has been used in previous

studies [27–40]. Large deformation and mesh distortion are challenges for the FEM analysis of deep anchors, especially in sand. Hakeem et al. used the arbitrary Lagrangian–Eulerian (ALE) method to simulate a circular plate anchor subjected to centric loading with $H/D = 1\sim20$ [11]. For deep anchors, the load–displacement curve was approximately a straight line with a $0.4D$ maximum displacement, and the $25\%D$ criterion was adopted to determine the ultimate uplift capacity; the authors proposed that this criterion is somewhat arbitrary.

Konkol et al. proposed that the ALE method is suitable for issues where soil displacement values are important, but the coupled Eulerian–Lagrangian (CEL) method is suitable for issues where mainly Lagrangian elements (piles, anchors, etc.) are analyzed [41]. Qiu et al. proposed that the CEL method can deal with large mesh distortions and contact problems [42]. It has proven successful in geotechnical problems such as a strip footing problem, installing a pile into the subsoil, and a ship running aground at an embankment [43,44]. In those studies, different constitutive models were used, such as the Mohr–Coulomb constitutive model, hardening soil model, and Drucker–Prager model. However, strain softening is important to incorporate into constitutive models of sand's behavior [45], so consideration of softening and shear swelling for large deformations may be more suitable for sand. Numerical results based on elastic–perfectly plastic or hardening soil models will overestimate the capacities of helical or circular anchors, especially in dense sand. Moreover, this method is widely used to study the load-bearing characteristics of plate or helical anchors in clay [42,44,46,47]. However, this method is very rarely used to study helical anchors in sand.

To solve the above issues, a numerical analysis was conducted, considering the impacts of embedment depths, helical plate spaces, and helical plate numbers. In this study, the coupled Eulerian–Lagrangian (CEL) approach in Abaqus/Explicit was employed for the large deformation analysis, combined with a modified Mohr–Coulomb model considering soil's strain softening and shear swelling, and the numerical model was validated by the centrifugal tests of the helical anchors (Section 2). Then, the deep failure model was presented according to the rupture surface observed via FEM (Section 3). Finally, an estimation of the uplift capacity was proposed by the limit equilibrium method, and the parameter sensitivity was studied (Section 4).

## 2. FEM Model and Validation

In the present study, the CEL approach in Abaqus was used to simulate the uplift behavior of helical anchors in sand, including the effects of density ($D_r = 30\%$, 60%, and 100%), embedment ratio ($H/D = 8\sim12$), helix spacing ratio ($S/D = 1.5\sim6$), and helical plate number ($n = 1\sim4$).

### 2.1. FEM Model

The FEM models were constructed as illustrated in Figure 3. Helical plates were replaced with circular plates based on previous findings that the plates' geometrical shape has little influence on the uplift capacity [6,12], and this analysis does not focus on the stress of the anchor body. Only one-quarter of the anchor and soil domain was considered in this study in terms of axisymmetry. Boundary conditions were imposed on the two planes of symmetry by prescribing zero flow velocity as normal to these planes. The bottom of the computational domain was constrained against the flow in the vertical direction. The single-helix anchors had an embedment ratio of 8~12. The lowermost plate of the multi-helix anchors had the same embedment ratio of 12, varied spacing ratios of 1.5~6.0, and varied plate numbers of 2~4.

For comparison with the results of the centrifugal test in dense sand [12], the helix diameter $D$, the helix thickness $t$, and the shaft diameter of anchor d were 400 mm, $0.05D$, and $0.235D$, respectively. The anchor was modeled as a discrete rigid solid part meshed with the eight-node linear brick, reducing the integration element C3D8R. A reference point was set on the top of the anchor, with constrained horizontal displacement and axial

rotation. The soil domain consisted of the eight-node reduced integration Eulerian element EC3D8R. A void layer with a 5D vertical distance above the ground was defined to allow the soil to heave and flow into the empty Eulerian elements during subsequent analysis. The computational domain size was $10D \times 2H$ (where $H$ is the embedment depth of the anchor), which is sufficiently large to ignore the far-field boundary effects [43]. The mesh was densified in the zone around the anchor, from 5D above the uppermost plate to 5D below the lowermost plate vertically, and 4D from the shaft centerline horizontally. The minimum element size ΔB was in the vicinity of the plate. The contact between the anchor and the soil was automatically identified. A general contact was adopted, with "hard contact" for the normal contact and a penalty contact method for tangential contact.



**Figure 3.** Numerical model. (**a**) Single-helix anchor. (**b**) Double-helix anchor.

Previously, the sand was represented using an elastoplastic Mohr–Coulomb constitutive model, which has limitations [48], and the sand softening was not reflected by the Mohr–Coulomb model in Abaqus. A modified Mohr–Coulomb model was used to consider the strain softening of sand, as shown in Figure 4.

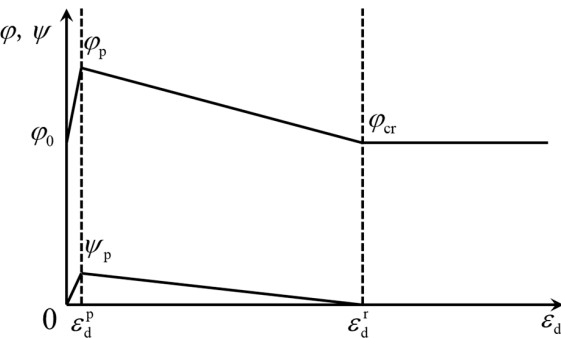

**Figure 4.** Diagrammatic sketch of the modified Mohr–Coulomb model.

The strength parameters, including the internal friction angle $\varphi$ and dilatancy angle $\psi$, were simplified to increase linearly to the peak values and then decrease linearly to the critical state values with the increase in equivalent plastic strain $\varepsilon_\mathrm{d}$, as shown in Equations (1) and (2), where the equivalent plastic strain can be calculated according to $\varepsilon = \frac{\sqrt{2}}{3}\left[(\varepsilon_1 - \varepsilon_2)^2 + (\varepsilon_2 - \varepsilon_3)^2 + (\varepsilon_3 - \varepsilon_1)^2\right]^{\frac{1}{2}}$.

$$\varphi = \begin{cases} \frac{\varphi_\mathrm{p} - \varphi_0}{\varepsilon_\mathrm{d}^\mathrm{p}}\varepsilon_\mathrm{d} + \varphi_0 & 0 \le \varepsilon_\mathrm{d} < \varepsilon_\mathrm{d}^\mathrm{p} \\ \frac{\varphi_\mathrm{cr} - \varphi_\mathrm{p}}{\varepsilon_\mathrm{d}^\mathrm{r} - \varepsilon_\mathrm{d}^\mathrm{p}}\left(\varepsilon_\mathrm{d} - \varepsilon_\mathrm{d}^\mathrm{r}\right) + \varphi_\mathrm{cr} & \varepsilon_\mathrm{d}^\mathrm{p} \le \varepsilon_\mathrm{d} < \varepsilon_\mathrm{d}^\mathrm{r} \\ \varphi_\mathrm{cr} & \varepsilon_\mathrm{d} > \varepsilon_\mathrm{d}^\mathrm{r} \end{cases} \quad (1)$$

$$\psi = \begin{cases} \frac{\psi_p}{\varepsilon_d^p}\varepsilon_d & 0 \le \varepsilon_d < \varepsilon_d^p \\ \frac{\psi_p}{\varepsilon_d^p-\varepsilon_d^r}\left(\varepsilon_d - \varepsilon_d^r\right) & \varepsilon_d^p \le \varepsilon_d < \varepsilon_d^r \\ 0 & \varepsilon_d > \varepsilon_d^r \end{cases} \tag{2}$$

The initial internal friction angle $\varphi_0$ is assumed to be the same as the critical internal friction angle $\varphi_{cr}$. The parameters contained in the model are Young's modulus $E$, Poisson's ratio $v$, the initial lateral pressure coefficient $K_0$, the peak internal friction angle $\varphi_p$, the peak dilatancy angle $\psi_p$, the critical internal friction angle $\varphi_{cr}$, the peak equivalent plastic strain $\varepsilon_d^p$, and the critical equivalent plastic strain $\varepsilon_d^r$.

The values of the modulus, peak friction angle, and dilatancy angle depend on confining pressure at the lowermost plate. For dense sand ($D_r = 100\%$) used in the centrifugal test [12], Young's modulus $E$ and the peak internal friction angle $\varphi_p$ are taken from the literature [49] after being obtained from triaxial tests; that is, $E = 658$ $p_a$ $(\sigma_3/p_a)^{0.469}$ and $\varphi_p = 40.9 - 7.8l$ g $(\sigma_3/p_a)$, where $p_a$ is the standard atmospheric pressure (101.3 kPa) and $\sigma_3$ is the confining pressure and takes the value of gravity stress $\gamma'H$, kPa. The values of $\varphi_p$ herein are relatively close to those for dense uwa sand in the recent literature [50,51], and their difference is 1~2°. The values of $E$ and $\varphi_p$ for loose sand ($D_r = 30\%$) and medium dense sand ($D_r = 60\%$) were calculated proportionally based on the literature [52]. The densities of different compactness were 1.61 g/cm³, 1.66 g/cm³, and 1.75 g/cm³, respectively. There was no groundwater influence, so the soil's effective unit weight $\gamma'$ was 15.8 kN/m³, 16.3 kN/m³, and 17.2 kN/m³, respectively. The peak dilatancy angle $\psi_p$ was estimated by the equation $\varphi_p = 0.5\psi_p + \varphi_{cr}$ proposed by Bolton [53]. The validity of the loose sand and medium dense sand parameters was determined by comparison with the triaxial test results in reference [49]. The soil parameters for different embedment ratios are summarized in Table 1. Other parameters were determined based on reference [49]—that is, $\varphi_{cr} = 31°$, $v = 0.3$, $K_0 = 1 - \sin\varphi_{cr} = 0.485$, $\varepsilon_d^p = 2\%$, and $\varepsilon_d^r = 20\%$.

**Table 1.** Soil parameters.

| $D_r$ | | 30% | | | 60% | | | 100% | |
|---|---|---|---|---|---|---|---|---|---|
| $H/D$ | $E$ (MPa) | $\varphi_p$ (°) | $\psi_p$ (°) | $E$ (MPa) | $\varphi_p$ (°) | $\psi_p$ (°) | $E$ (MPa) | $\varphi_p$ (°) | $\psi_p$ (°) |
| 8 | 24.27 | 34.77 | 7.54 | 35.47 | 38.31 | 12.26 | 50.4 | 42.9 | 23.8 |
| 9 | 25.64 | 34.65 | 7.3 | 37.48 | 38.07 | 11.86 | 53.26 | 42.5 | 23 |
| 10 | 26.94 | 34.54 | 7.08 | 39.38 | 37.86 | 11.5 | 55.96 | 42.14 | 22.28 |
| 10.5 | 27.57 | 34.49 | 6.98 | 40.29 | 37.76 | 11.34 | 57.25 | 41.98 | 21.95 |
| 12 | 29.35 | 34.36 | 6.71 | 42.89 | 37.49 | 10.89 | 60.95 | 41.52 | 21.05 |

### 2.2. Influence of Pullout Rate and Mesh Density

The pullout process of a plate anchor is essentially quasi-static in nature, while the Eulerian analysis is formulated in the framework of a dynamic explicit solution scheme instead of a static implicit framework [43]. To achieve a balance between matching the quasi-static state as closely as possible and reducing the computational time, a parametric study was carried out to investigate the effect of the pullout rate. Three pullout rates of 0.025 $D$/s, 0.05 $D$/s, and 0.1 $D$/s were considered, with a minimum element size $\Delta B = 0.1D$. Also, a mesh convergence study was performed to identify a suitable mesh density that gives sufficiently accurate results. Both studies were performed in the case of a single-helix anchor with $H/D = 9$ in dense sand.

The load–displacement curves of different pullout rates are illustrated in Figure 5a, with the minimum element size $\Delta B = 0.1D$. The pullout rate V has little influence on the uplift process. When the displacement is up to 0.5$D$, the computational time of $V_1$ and

$V_2$ is about 12 times and 2 times that of $V_3$, respectively. Hence, considering the time consumption and stability, the pullout rate $V_2$ was adopted for all subsequent analyses.

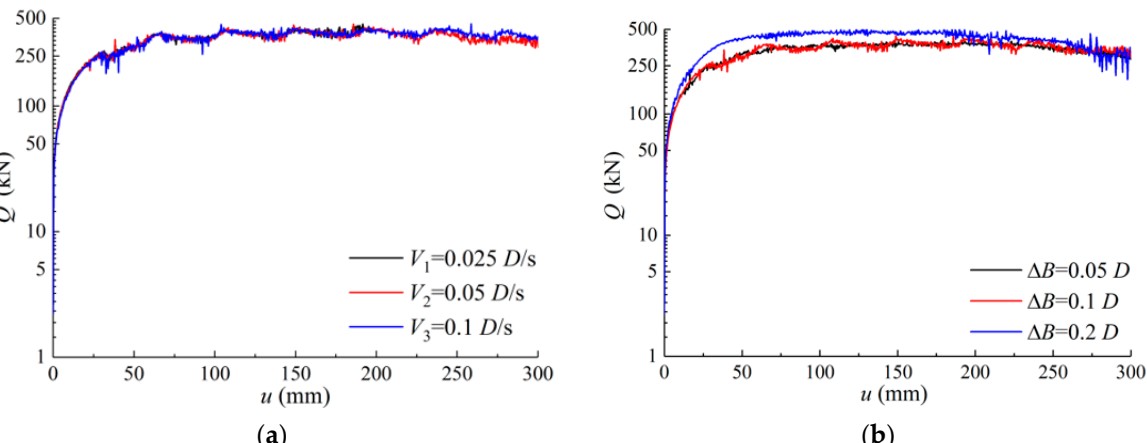

**Figure 5.** The curves of uplift resistance and displacement for the convergence study. (**a**) Pullout rate study. (**b**) Mesh convergence study.

Three finite element meshes with minimum element sizes $\Delta B$ near the anchor of 0.05$D$, 0.1$D$, and 0.2$D$ were compared, and the element numbers were 613130, 280864, and 125856, respectively, where the pullout rate $V = 0.05\ D/s$. The load–displacement curves of different mesh densities are illustrated in Figure 5b. It can be seen that a finer mesh tends to give a smaller uplift capacity, and that the mesh with the minimum size $\Delta B = 0.1D$ is preferable because a further decrease in mesh size does not change the result. Therefore, meshing with a minimum element size of 0.1$D$ was considered sufficient in terms of accuracy and was adopted for all subsequent analyses.

### 2.3. Determination and Verification of Uplift Capacity

The relationship between load and displacement of single-helix anchors ($H/D = 8 \sim 12$) in sand with different densities is presented in Figure 6. The relationship between load and displacement of multi-helix anchors in dense sand is shown in Figure 7. The characteristics of these curves can be divided into two types: those with an obvious peak point, and those without an obvious peak point. The peak values of the curves were undoubtedly taken as the ultimate uplift capacities, and the uplift capacities for the curves without obvious peak points were determined according to the curve development. The development of these curves presents three stages: the fast-rising straight-line stage, the curved stage, and a prolonged rising or stable stage (approximately a straight line). The starting point of the latter stationary section was regarded as the ultimate uplift capacity, $Q_u$, and the corresponding displacement is called failure displacement, uf. The points of $Q_u$ are represented by circles in Figures 6 and 7.

The uplift capacity is normalized as the uplift capacity factor $N_\gamma$, as in Equation (3):

$$N_\gamma = Q_u / \gamma' A H \tag{3}$$

where $\gamma'$ is the soil's effective unit weight, and $A$ is the plate area, expressed as $A = \pi D^2 / 4$.

The uplift capacity factors $N_\gamma$ for single-helix anchors from the FEM and centrifugal tests are plotted in Figure 8a, which shows that the uplift capacity factors of single-helix anchors are roughly constant as the embedment ratio $H/D$ increases in loose sand, medium dense sand, and dense sand with $H/D > 9$. This feature is related to the failure mode, which will be explained later. The uplift capacity factors from the centrifugal test with $D_r = 85.4 \sim 96.2\%$ are encompassed between the numerical results of medium and dense sand. This result verifies the reliability of the FEM model. However, the FEM results overestimated the uplift capacities of helical anchors. This difference may be caused by

the dilatancy angle determined by Bolton's equation [53], which is larger than the actual situation.

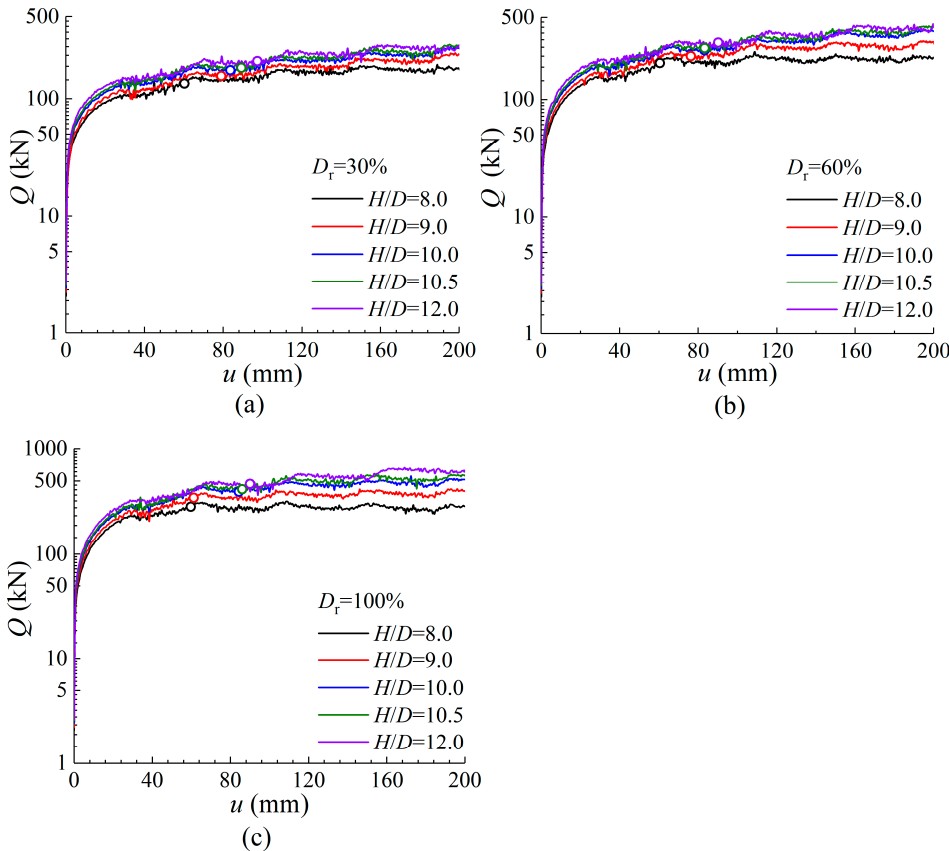

**Figure 6.** The relationship between load and displacement of single-helix anchors. (**a**) Loose and. (**b**) Medium dense sand. (**c**) Dense sand.

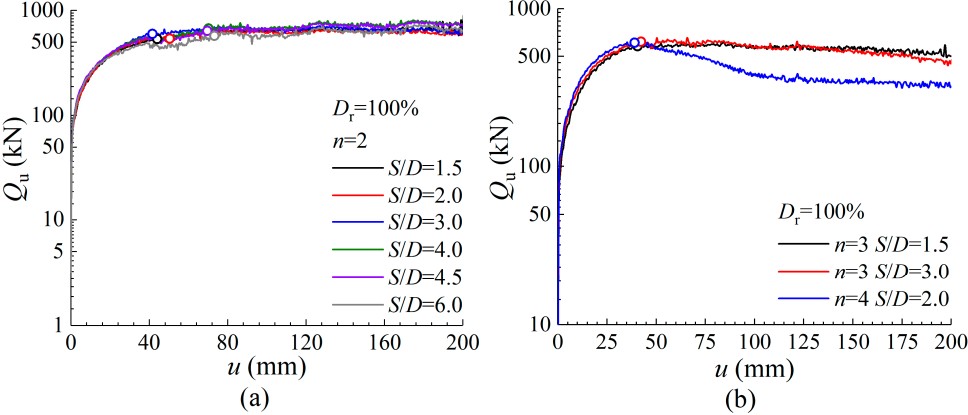

**Figure 7.** The relationship between load and displacement of multi-helix anchors. (**a**) Double-helix anchor. (**b**) Multi-helix anchor.

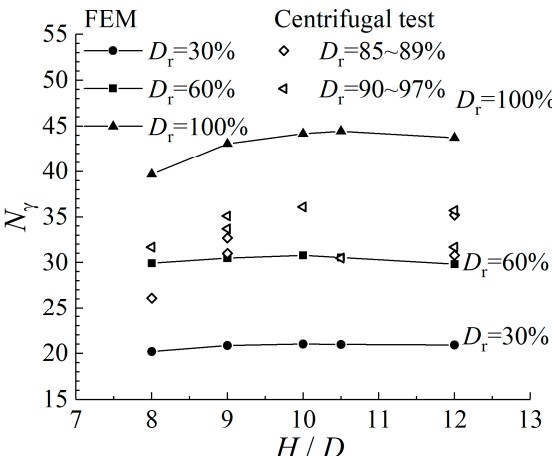

**Figure 8.** The comparison between the numerical and centrifugal test results.

## 3. Uplift Failure Mode

The effect of strain softening was incorporated into numerical analysis by utilizing the modified Mohr–Coulomb model described in Section 2, which allows for the variation of soil strength with strain and stress level and can show the subsequent soil strength mobilization and progressive failure. The failure surface was identified based on the contour of equivalent plastic strain at failure displacement, and the design method of the deep helical anchor is further proposed based on the visible deep failure mode in Section 4.

### 3.1. Process of Soil Strength Mobilization

Figure 9 shows the process of soil strength being mobilized while pulling the anchors, where $u_f$ is the displacement corresponding to the ultimate uplift capacity defined above. The red zone corresponds to the peak internal friction angle, and the blue zone corresponds to the initial internal friction angle or the critical internal friction angle. The soil inside the red zone is undergoing softening, and the soil outside the red zone is experiencing hardening. The soil strength near the anchor plate is mobilized first and reaches its peak state. Then, the strength of this part of the soil reduces with the increase in the displacement or strain and reaches a critical state. The peak state boundary expands successively, and all of the soil within this boundary undergoes a softening process. This reflects the process of progressive failure of the foundation.

The local failure mode is present for different densities. As the displacement $u$ increases, the influence range of the interaction between the anchor plate and soil expands gradually, and the soil strength of the partial zone around the plate is mobilized. The mobilized zone still expands outward gradually after the failure displacement $u_f$. Although a larger range of soil strength is mobilized, the soil strength near the plate decreases after peak strength. Hence, the overall uplift capacity shows a trend of slow increase, which corresponds to the load–displacement curve shown in Figure 6. The shapes of the mobilized zone in the deep underground area at failure displacement for different densities are similar, but the range becomes larger with the higher density. The very shallow soil near the ground surface is also mobilized due to shaft friction and low overburden pressure, which are not features of failure mode recognition.

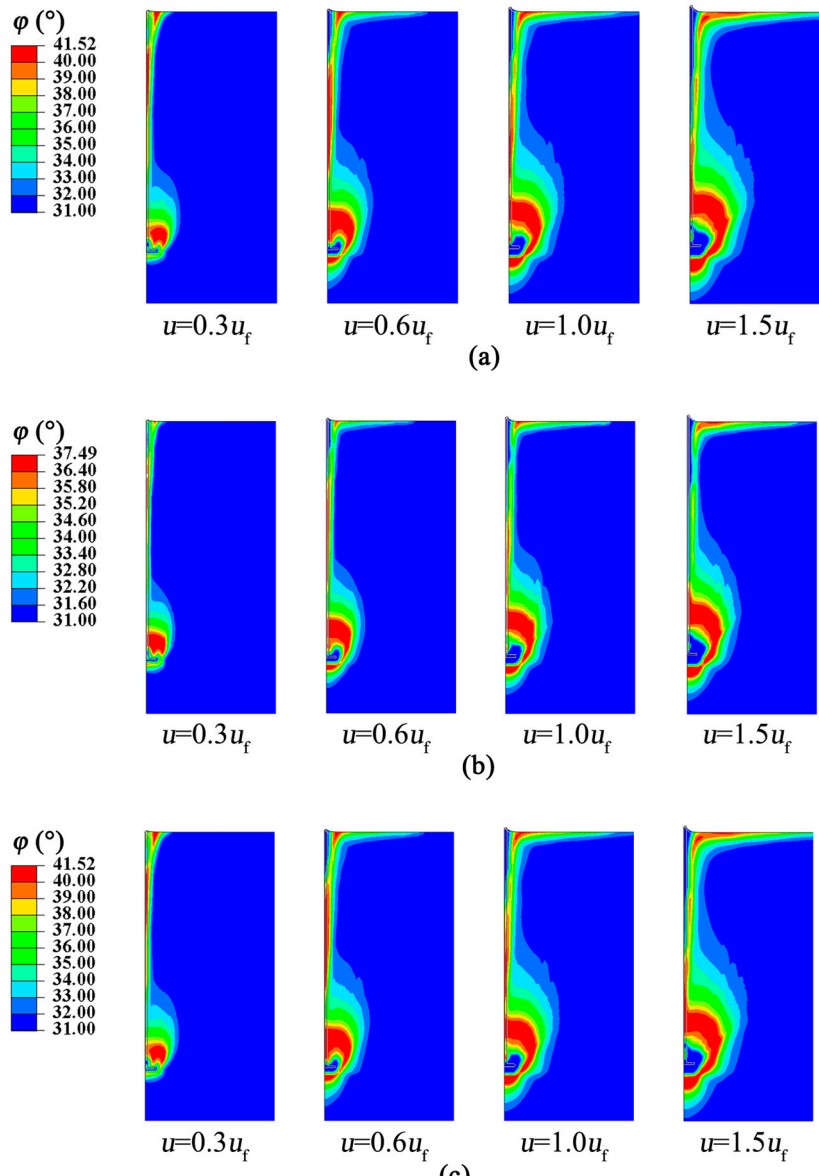

**Figure 9.** Contours of internal friction angle at different uplift displacements ($H/D$ = 12). (**a**) Loose sand. (**b**) Medium dense sand. (**c**) Dense sand.

### 3.2. Deep Failure Behavior of Single-Helix Anchors

The equivalent plastic strain contours of the single-helix anchors at failure, with different embedment ratios $H/D$ and relative densities, are shown in Figure 10. The white zone corresponds to the immobilized zone; the black zone corresponds to the softening zone, with an equivalent plastic strain greater than 2%; and the other zone corresponds to the hardening zone. The plastic zone surrounding the plate is limited below the ground surface and displays a closed bulb, which is the deep failure mode. However, the failure mode for the anchor with an embedment ratio of eight in dense sand displays the transitional mode, which will not be discussed in the following text.

For the deep failure mode, the influence of density on the mobilized zone range is greater than that of the embedment ratio. The boundary of the mobilized zone increases gradually with the increase in soil density. Its vertical height is 5$D$, 6$D$, and 7$D$, and the horizontal breadth is 2$D$, 2.5$D$, and 3$D$, in the loose, medium, and dense sand, respectively. However, the scope of the softening zone is hardly affected by density and embedment ratio. Its vertical height is approximately 2$D$, and the horizontal breadth is approximately

1.5*D* for different densities and embedment ratios. Due to the similarity of the failure modes, the dimensionless uplift capacities of deep single-helix anchors are approximately constant.

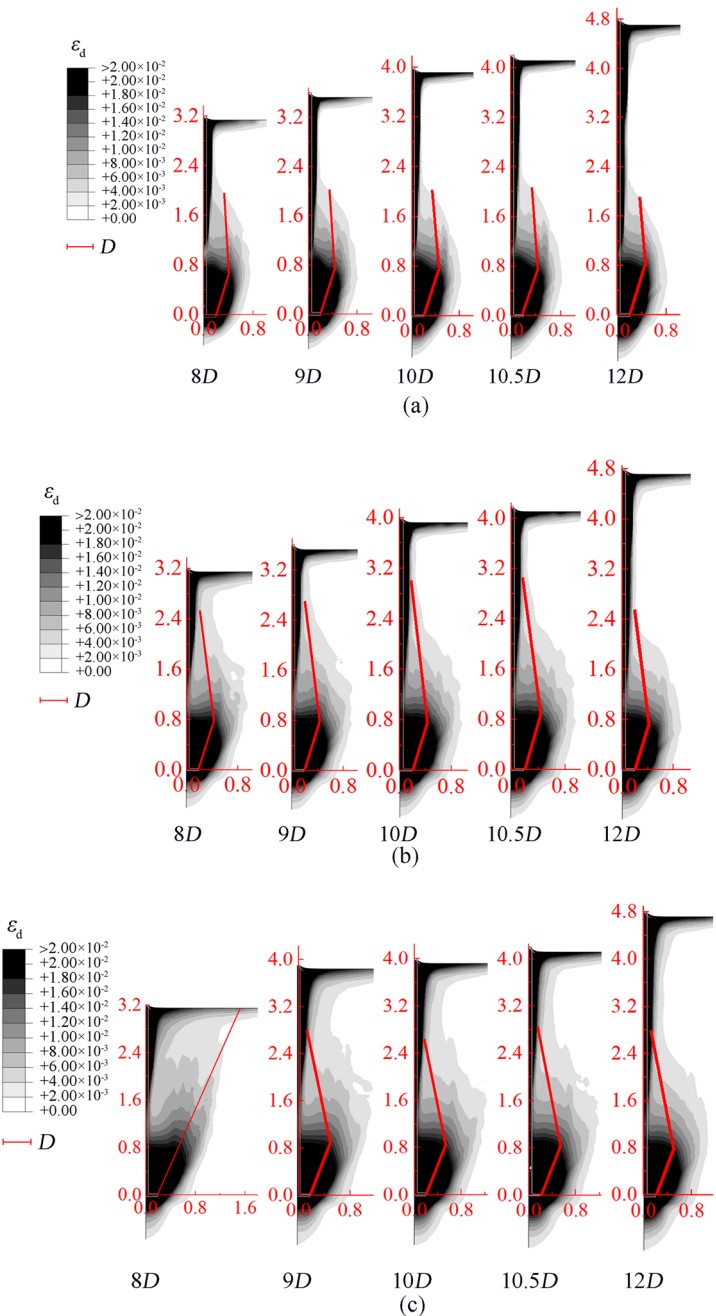

**Figure 10.** Single-helix anchor cases: contour of equivalent plastic strain and rupture surface (red line). (**a**) Loose sand. (**b**) Medium dense sand. (**c**) Dense sand.

For the convenience of calculation, Ghaly [13] assumed the deep failure mode as an inverted cone with a certain height. This assumption is similar to the experimental results observed by Motamedinia [18] and Salehzadeh [19]. The overburden pressure was simplified as a uniform load acting on the top of the inverted cone. Because the top of the inverted cone has a larger surface area than the potential region for load, this assumption can lead to excessive earth pressure. Liu [54] and Shi [55] proposed that the rupture surface can be determined by the points of maximum strain at varied depths based on the contour of plastic shear strain. This numerical analysis adopts the rule to identify the rupture

surfaces of deep single-helix and multi-helix anchors. The deep rupture surface of the single-helix anchor can be simplified as two straight lines, as shown by the solid red line in Figure 10. From the plate edge to the boundary of the softening zone, there is an inverted truncated cone. The vertical distance between them is within (2~3)$D$. The inclination in the vertical direction rises from 15° to 20°. Then, an erected truncated cone extends from the boundary to the shaft. The vertical distance between them is within (4~6)$D$. The inclination in the vertical direction rises from 4° to 10°.

### 3.3. Deep Failure Behavior of Multi-Helix Anchors

Figure 11 shows the equivalent plastic strain at failure of multi-helix anchors buried in dense sand with an embedment ratio of 12$D$ and various spacing ratios $S/D$ and plate numbers $n$. It can be seen that the embedment depth of the uppermost plate controls the failure mode of the multi-helix anchor. Although the lowermost plate of all of these anchors is deeply embedded (12$D$), the mobilized zone of soil around the anchors extends to the ground, except for the double-helix anchors with $S/D$ = 1.5~3 and the triple-helix anchor with $S/D$ = 1.5. The softening zones are interconnected when the plate spacing is smaller than 6$D$. This failure mode is similar to the one assumed by the cylindrical shear method.

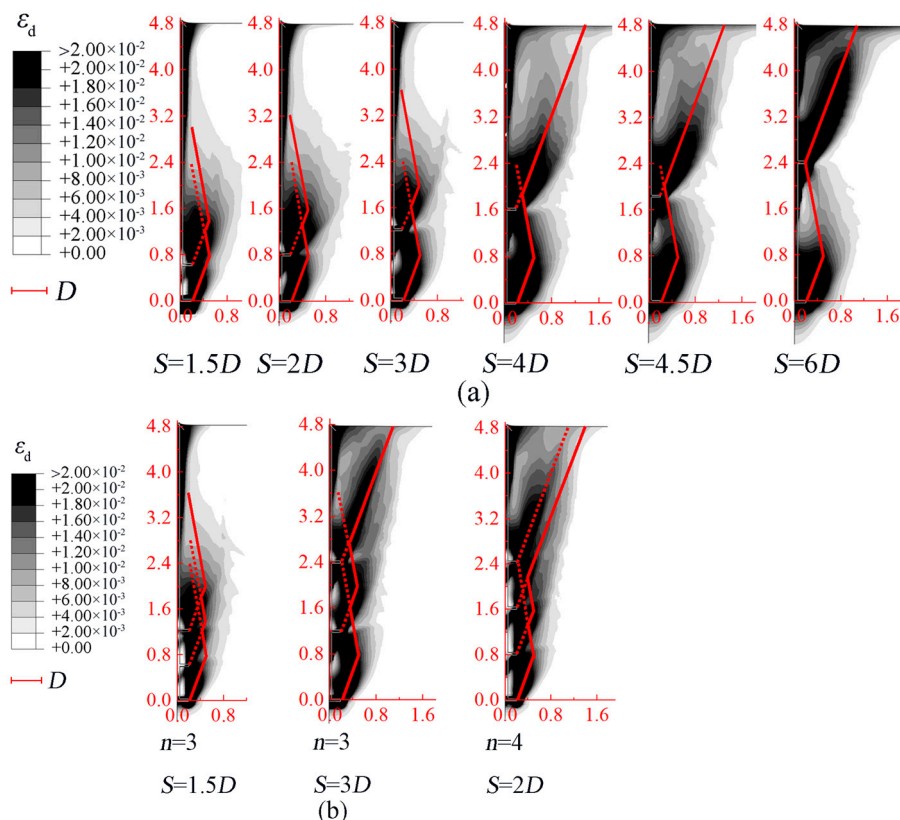

**Figure 11.** Multi-helix anchor cases: contour of equivalent plastic strain and rupture surface (red line) (dense sand). (**a**) Double-helix anchor. (**b**) Multi-helix anchor.

Compared with the failure mode of the single-helix anchor shown in Figure 11, it is worth noting that the helix spacing has little influence on the equivalent plastic strain range induced by each helical plate. For example, the soil mobilized zone of the double-helix anchor with $S/D$ = 2 is just like a superposition of two single-helix anchors with $H/D$ = 12 and $H/D$ = 9. The upper plate experiences a transitional failure mode when the $S/D$ = 4 for a double-helix anchor. When the $S/D$ = 6 for the double-helix anchor, it is evident that the boundary of the soil mobilized zone caused by the lower plate bends towards the shaft. For multi-helix anchors, if more than one plate is at a shallow depth, the soil mobilized zone caused by the lowermost shallow plate will cover the mobilized zone of the other

shallow plates, as with the quadruple-helix anchor shown in Figure 11b. The numerical observation confirms the conclusion from the centrifugal test that additional helices do not provide additional capacity if they are located within the region of soil mobilized by a lower helix [12]. As a result, the uplift resistance of the multi-helix anchor should be provided by the lowest shallow plate and each deep plate. The range of the soil plastic zone caused by each plate is only tied to the plate's embedment depth. The rupture surface of the multi-helix anchor can be regarded as the superposition of each helical plate's rupture surface when the above rule is adopted to identify the rupture surfaces [54,55], as shown by the solid red line in Figure 11.

## 4. Estimation of Uplift Capacity

It was possible to establish the link between the rupture surface and soil parameters, because varied soil characteristics were used. A simplified rupture surface was put forward according to the failure mode (Section 3). A unified calculation method using the limit equilibrium method (LEM) was proposed, including the lateral earth pressure coefficient, average internal friction angle, exponential decrease rate, and other parameters. The meanings of each symbol can be found in the Appendix A.

### 4.1. Simplified Rupture Surfaces

According to the failure mode determined from the above numerical results, the shallow and transition rupture surfaces of a single-helix anchor can be represented by one inverted truncated cone (inclined at $\psi_p$ to the vertical), as illustrated in Figure 12a, which is the same as in previous experimental investigations [14,18,56,57]. The deep rupture surface of the single-helix anchor can be represented by one erected and one inverted truncated cone, as illustrated in Figure 12b. The inverted cone emerges from the plate edge with a vertical height of $2D$ and an inclination to the vertical of $\varphi_p/2$, and connected to the erected cone with a vertical height of $4D$ and an inclination to the vertical of $\psi_p/2$.

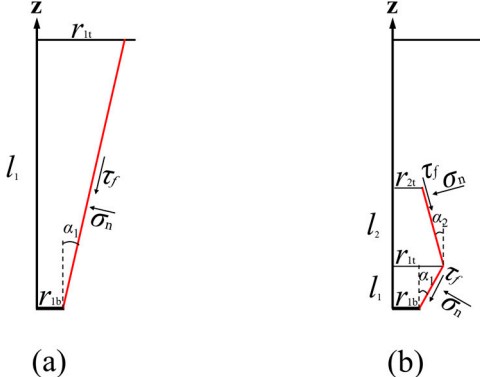

(a)            (b)

**Figure 12.** The simplified rupture surface of the single-helix anchor. (**a**) Shallow mode. (**b**) Deep mode.

The double-helix anchor is taken as an example. The deep rupture surface can be represented by two erected and two inverted truncated cones. If the rupture surface of the adjacent plate does not intersect, each plate can be calculated individually, as shown in Figure 13a. If the rupture surface of an adjacent plate intersects, its outermost contour is taken as the rupture surface, as shown in Figure 13b. Figure 13a,b cover the possible shallow and deep failure modes of the top helical plate simultaneously. The same simplification method can be applied to the multi-helix anchor. Thus, the rupture surface can be divided into several erected and inverted truncated cones. It is worth noting that the additional helical plates do not provide additional capacity if they are located within the region of soil mobilized by a lower plate.

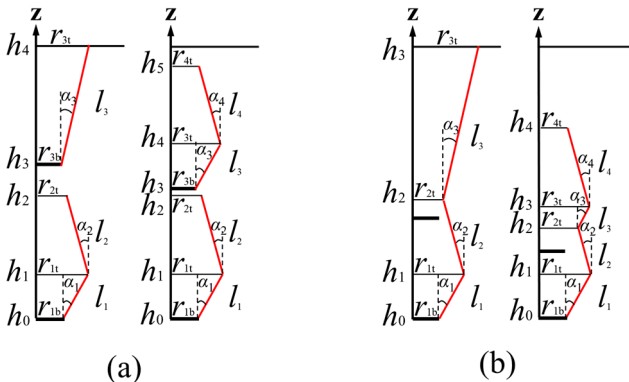

**Figure 13.** The simplified rupture surface of the double-helix anchor. (**a**) Individual mode. (**b**) Intersection mode.

### 4.2. A Unified Calculation Method

According to the limit equilibrium method, the ultimate uplift capacity of a single-helix anchor $Q_u$ equals the sum of the vertical component of shearing resistance along each truncated cone and the soil weight, as shown in Equation (4):

$$Q_u = \sum (Q_{ui} + W_i) \tag{4}$$

where $Q_{ui}$ is the uplift capacity of the $i$th truncated cone, as calculated using Equation (5):

$$Q_{ui} = \int_A \left( \tau_f \cos \alpha_i + (-1)^i \sigma_n \sin \alpha_i \right) \mathrm{d}A \tag{5}$$

where $\sigma_n$ is the normal stress on the rupture surface, $\sigma_n = K_u \gamma'(H - z)$, $K_u$ is the lateral earth pressure coefficient, $z$ is the vertical distance from the lowermost plate, $H$ is the embedment depth of the plate, $\tau_f$ is the resisting shear stress on the rupture surface, $\tau_f = \sigma_n \tan \varphi^*_A$, and $\varphi^*_A$ is the average internal friction angle. $K_u$ and $\varphi^*_A$ are determined as shown in the following section. The inclination angle to the vertical $\alpha_i$ for the deep plate is $\varphi_p/2$ when $i$ is an odd number, and $\alpha_i = \psi_p/2$ when $i$ is an even number. The shallow plate is assumed to be the inverted truncated cone, whose inclination angle is $\psi_p$.

$W_i$ is the soil weight in the $i$th rupture surface of the truncated cone, as calculated using Equation (6):

$$W_i = \frac{\pi}{3} \gamma' l_i \left( r_{it}^2 + r_{ib}^2 + r_{it} r_{ib} \right) \tag{6}$$

where $l_i$ is the vertical height of the $i$th rupture surface, and its bottom and top radii are $r_{ib}$ and $r_{it}$, respectively. These parameters can be determined by geometry.

The above equations are also applied to the multi-helix anchor, since its failure mode is the superposition of the failure modes of each helical plate. Because the multi-helix anchor's failure mode is the superposition of each helical plate's failure mode, only the lateral earth pressure coefficient and the average internal friction angle of the single-helix anchor will be discussed.

### 4.3. Lateral Earth Pressure Coefficient

The effect of soil deformation will change the stress field around the plate, which is the significant difference between FEM and limit analysis. The stress distribution along the rupture surface of shallow circular anchors in sand was reflected by Cerfontaine [8] using FEM based on the hardening soil constitutive model, which can be described by a linearly increasing and then exponentially decreasing mathematical function. However, the stress analysis of deep anchors in sand is still unclear. This study investigated the stress distribution along the rupture surface of a deep anchor based on a modified Mohr–Coulomb model that incorporates the mobilized process of soil strength with plastic strain.

According to the Mohr–Coulomb limit equilibrium condition, the normal stress $\sigma_n$ can be calculated by the maximum and minimum principal stresses and the internal friction angle of each element along the rupture surface. Then, the lateral earth pressure coefficient of deep anchor $K_u$ along the rupture surface can be calculated by Equation (7):

$$K_u = \sigma_n / \gamma'(H - z) \tag{7}$$

where $z$ is the vertical distance from the lowermost plate.

For shallow single-helix anchors, the lateral earth pressure coefficient $K_n$ was proposed by Hao et al. [12], based on the assumption that the normal stress on the rupture surface remains in its initial state during pullout, as shown in Equation (8):

$$K_n = 1 - \frac{\sin \varphi_{cr}(1 + \cos 2\psi_p)}{2} \tag{8}$$

Then, $K_u$ was standardized with $K_n$, and the distributions of $K_u / K_n$ for different densities are shown in Figure 14.

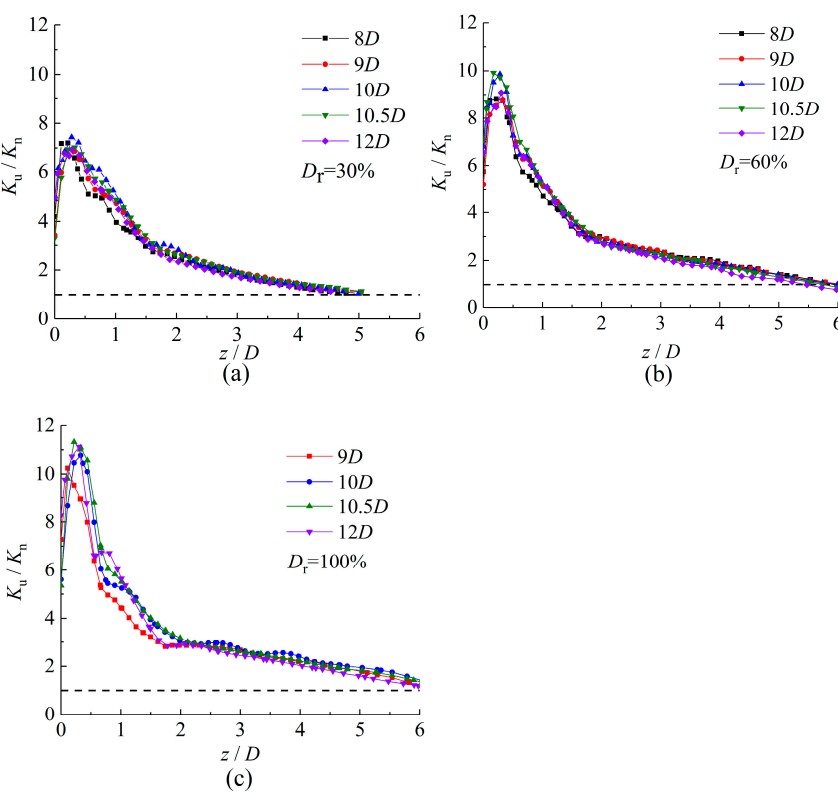

**Figure 14.** Distribution of normalized coefficients of lateral earth pressure along rupture surface. (**a**) Loose sand. (**b**) Medium dense sand. (**c**) Dense sand.

It can be seen from Figure 14 that the value of $K_u$ increases linearly to the peak value $K_{u,peak}$ rapidly, and then it slowly decreases to close to the value of $K_u$. The trend of stress distribution is similar to the stress investigation around helical piles using the photoelasticity technique reported by Schiavon [16]. It is worth noting that the density has a more significant effect on the peak $K_{u,peak}$ instead of the embedment depth.

Considering the trend of $K_u$, two-stage functions are used to express the variation of $K_u$, as shown in Figure 15. Linear fitting is adopted before the peak value, which starts from the initial lateral pressure coefficient $K_n$ to the peak lateral pressure coefficient $K_{u,peak}$. Then, exponential fitting is adopted after the peak value, which starts from the initial lateral

pressure coefficient $K_{u,peak}$ to the lateral earth pressure coefficient proposed Hao [12] $K_n$, as shown in Equation (9):

$$K_u = \begin{cases} K_n + \left( K_{u,peak} - K_n \right) / z_p \cdot z & z \leq z_p \\ K_n + \left( K_{u,peak} - K_n \right) e^{-\kappa(z-z_p)} & z > z_p \end{cases} \tag{9}$$

where $z_p$ is the vertical distance of the peak value point from the plate, and $\kappa$ is the exponential decrease rate. The position of $z_p$ is always within $0.33D$ and has little relation to the relative density and embedment depth. Thus, $z_p$ can be taken as $0.33D$ in Equation (9). The value of $K_{u,peak}$ in loose, medium, and dense sand is 3.5, 5, and 6.5, respectively. The value of $\kappa$ is fitted to be 1.8.

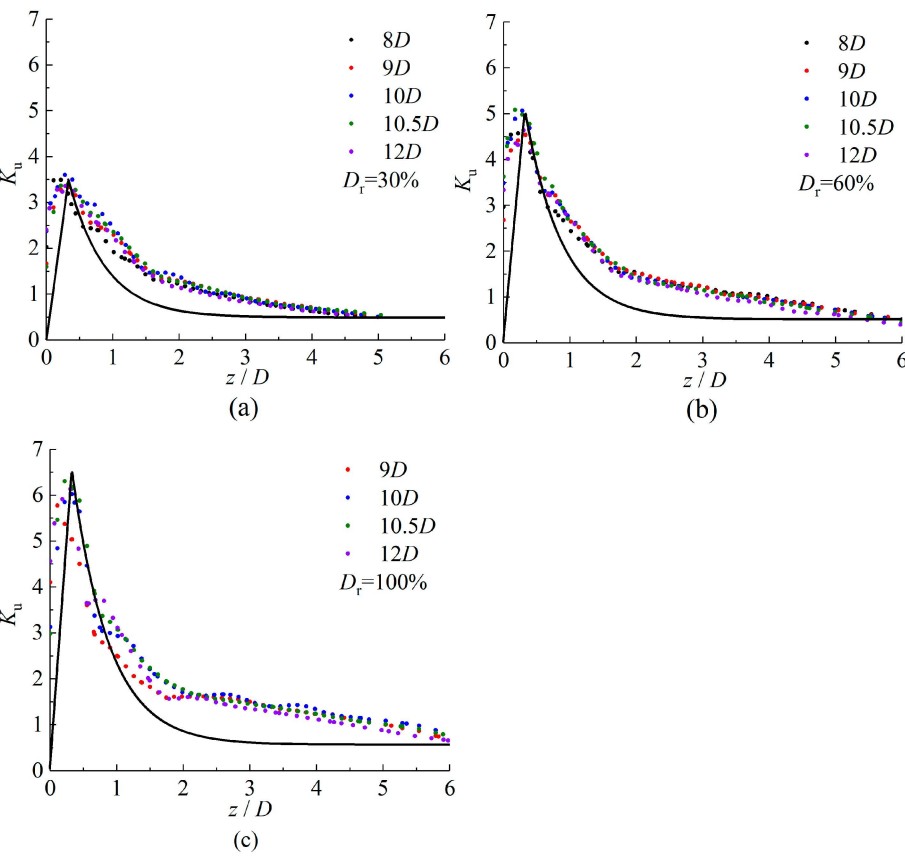

**Figure 15.** Simplified lateral earth pressure coefficient $K_u$. (**a**) Loose sand. (**b**) Medium dense sand. (**c**) Dense sand.

To simplify Equation (9), $z_p$ can be taken as 0, because the value of $z_p$ is small. The exponential fitting is adopted to express the variation of $K_u$, as shown in Equation (10). This part will be discussed in Section 4.5.

$$K_u = K_n + \left( K_{u,peak} - K_n \right) e^{-\kappa z} \tag{10}$$

### 4.4. Average Internal Friction Angle

According to the modified Mohr–Coulomb model, the internal friction angle is related to the equivalent plastic strain. Its distribution along the rupture surface is shown in Figure 16. The peak value point is vertically located $2D$ away from the plate, which corresponds to the boundary of the softening zone and is different from the position of the peak lateral earth pressure coefficient.

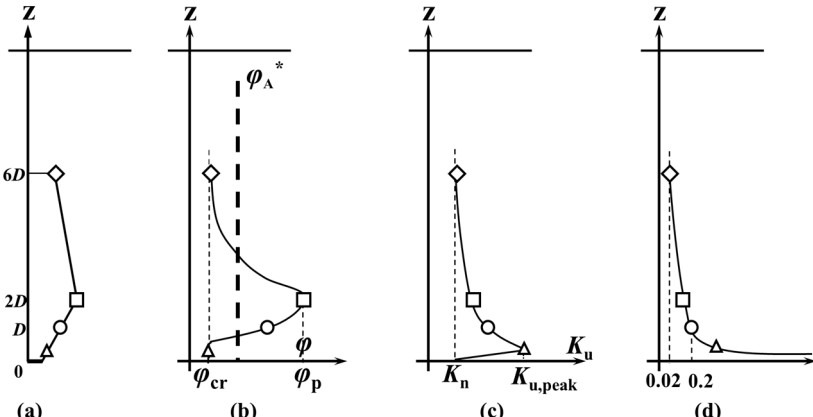

**Figure 16.** Distribution of internal friction angle, lateral earth pressure coefficient, and equivalent plastic strain along the rupture surface. (**a**) Rupture surface. (**b**) Internal friction angle. (**c**) Lateral earth pressure coefficient. (**d**) Equivalent plastic strain.

The internal friction angle near the plate equals the critical friction angle. As the distance from the plate increases, it begins to rise until it reaches the peak, and then it gradually decreases. This tendency makes the analysis more complex and realistic than FEM based on the HS small model [8]. The shear stress along the failure surface varies not only with the effective normal stress but also with the mobilization level of soil strength, which depends on plastic strain. Drescher [58] pointed out that numerical solutions to the bearing capacity problem at $\psi < \varphi$ predict a limit load higher than the estimates presented for Prandtl's failure mechanism. Davis [59–61] proposed Equation (11) to convert a non-associated plastic MC model into an equivalent associated plastic MC model [61]:

$$\tan \varphi^* = \frac{\sin \varphi_{\mathrm{P}} \cos \psi_{\mathrm{P}}}{1 - \sin \varphi_{\mathrm{P}} \sin \psi_{\mathrm{P}}} \tag{11}$$

where $\varphi^*$ is the internal friction angle of the equivalent associated plastic MC model.

For the convenience of calculation and application in LEM, an average internal friction angle $\varphi^*_{\mathrm{A}}$ was proposed. The internal friction angle along the rupture surface was extracted, and the average value was calculated. The comparison of $\varphi^*_{\mathrm{A}}$ from FEM with $\varphi^*$ from Equation (11) [59] is shown in Figure 17. It can be seen that both values are similar for loose and medium dense sand, and the value of $\varphi^*$ is 3° greater than that of $\varphi^*_{\mathrm{A}}$ for dense sand. For simplification and safety, the average friction angle can be estimated by Equation (11) directly for loose and medium dense sand, and by reducing the results of Equation (11) by 3° for dense sand.

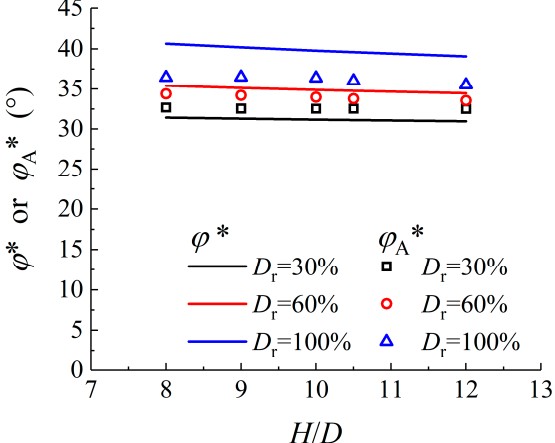

**Figure 17.** Comparison of the average internal friction angles from FEM and the calculation formula.

### 4.5. Comparison with Results

In Sections 4.1–4.4, the simplified rupture surfaces and the unified calculation method for helical anchors were discussed, and two important parameters were studied. So, the ultimate uplift capacity of a helical anchor can be estimated using Equations (4)–(11).

(1)     Comparison with the FEM results

To verify the feasibility of the simplified rupture surface, the calculation results were compared with the finite element results, and the soil pressure coefficient was calculated using Equation (9). When the top anchor plate is a shallow embedment plate for multi-helix anchors, the lateral earth pressure coefficient is determined using Equation (8) without considering its changes. The comparison results are shown in Figure 18. The uplift capacity predicted by this method is about 5% lower on average than that of the finite element results. But for single helical anchors, the result with the largest deviation is found for $H/D = 9$ in loose sand, which is underestimated by up to 8.3%. For double-helix anchors, although it corresponds nicely when the spacing is bigger, it is 16.1% underestimated when the spacing is $3D$. As a result, this demonstrates that it is possible to calculate the uplift capacity of helical anchors using the simplified rupture mode presented in Section 4.1.

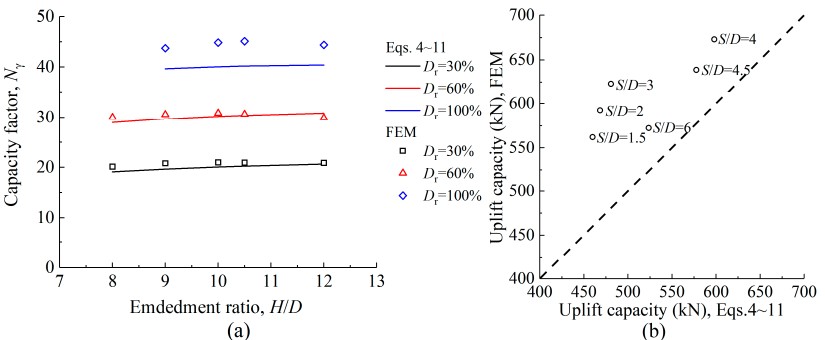

**Figure 18.** Comparison between the theoretical results and the FEM results. (**a**) Single-helix anchor. (**b**) Double-helix anchor.

(2)     Comparison between Equations (9) and (10)

To verify the feasibility of the simplified calculation method, the uplift capacity of single-helix anchors was calculated using Equations (9) and (10) as the lateral earth pressure coefficient. The results are shown in Figure 19. It can be seen that the simplified calculation results using Equation (10) will reduce the estimated uplift capacity. The values of $N_\gamma$ for loose sand, medium dense sand, and dense sand are reduced by an average of 7.5%, 9.3%, and 10.5%, respectively. However, for safety reasons, Equation (10) can be used instead of Equation (9) for calculation.

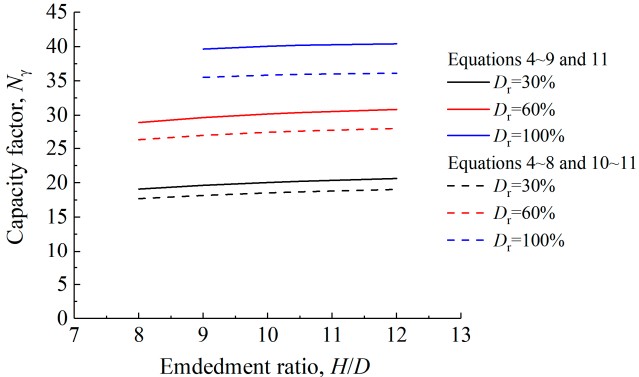

**Figure 19.** Comparison between Equations (9) and (10).

(3)    Comparison with the test results

To verify the unified calculation method, the calculation results were compared with the centrifuge test results [12]. The soil characteristics were based on the values in Section 2.1, and the lateral earth pressure coefficient was calculated using Equation (10). When the top anchor plate is a shallow embedment plate for multi-helix anchors, the lateral earth pressure coefficient is determined using Equation (8) without considering its changes. The results of this theoretical method were compared with the test results, as shown in Figure 20. It can be seen that for single-helix anchors in dense sand, this theoretical method underestimates by up to 10.9% and overestimates by up to 6.7%. For multi-helix anchors in dense sand, this method underestimates by up to 17.0% and overestimates by up to 21.7%. Therefore, this method may be used to estimate the uplift capacity of helical anchors.

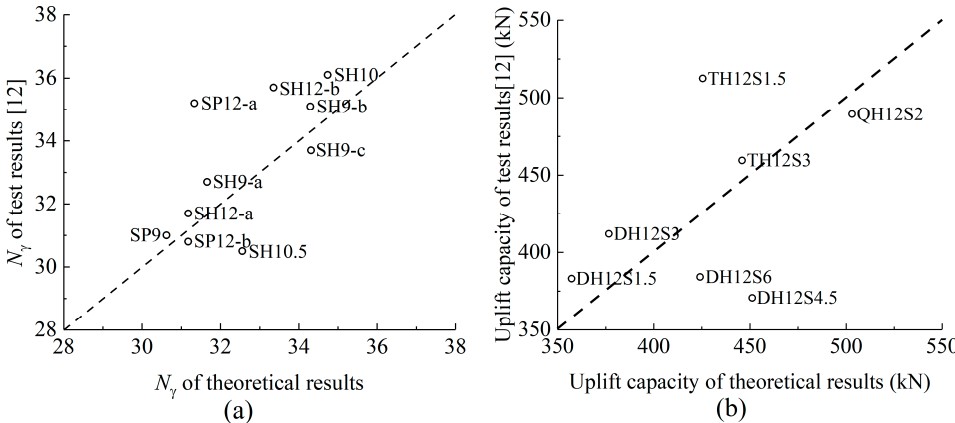

**Figure 20.** Comparison between the theoretical results and the centrifuge test results. (**a**) Single-helix anchor. (**b**) Multi-helix anchor.

## 5. Conclusions

The failure mechanism of deep anchors in sand is not well understood. The existing design methods of deep helical anchors in sand seldom consider the deep rupture surface. This study shows the possibility of simulating the pullout process of deep helical anchors in sand. Considering the strain softening of soil can help to more accurately simulate the uplift characteristics of deep helical anchors in sand, which can reflect the progressive failure process of deep anchors. In this process, the boundary of the plastic zone is a closed bulb and expands rapidly before the failure displacement and slowly after the failure displacement. A two-truncated-cone failure mode was identified based on the locus of maximum value of plastic strain at various depths for deep single-helix anchors, and on this basis, the analytical method of estimation for the ultimate uplift capacity of deep helical anchors is proposed.

The helix spacing has little influence on the equivalent plastic strain range induced by each helical plate. The boundary of the plastic zone for multi-helix anchors is a superposition of each helical plate. The distributions of the lateral earth pressure coefficient $K_u$ can be simplified by a linearly increasing and then exponentially decreasing mathematical function. The peak value of $K_u$ and the exponential decrease rate $\kappa$ are almost independent of the embedment ratio of deep anchors, but they are related to the sand's compactness. The value of the average internal friction angle can be determined by the Davis formula.

This study proposes a unified calculation method for helical anchors in sand, and the parameters in this method were determined according to the results of numerical analysis. This innovative method can be suitable for single-helix or multi-helix anchors, as well as shallow- or deep-embedment helix anchors. It may be helpful in the design of helical anchors. These findings could improve the current design method of helical anchors, especially deep-embedment helical anchors.

**Author Contributions:** Conceptualization, formal analysis, and writing—review and editing, D.H. and C.Y.; investigation and writing—original draft preparation, C.Y. and N.Z.; investigation and data curation, C.Y. and R.C. All authors have read and agreed to the published version of the manuscript.

**Funding:** This research was funded by the National Natural Science Foundation of China (grant number 52078108).

**Institutional Review Board Statement:** Not applicable.

**Informed Consent Statement:** Not applicable.

**Data Availability Statement:** Due to the nature of this research, participants of this study did not agree for their data to be shared publicly, so supporting data is not available.

**Conflicts of Interest:** The authors declare no conflict of interest.

## Appendix A

**Table A1.** Parameters in the paper.

| Parameters | Definition |
|---|---|
| $D$ | Helix diameter |
| $t$ | Helix thickness |
| $H$ | Embedment |
| $S$ | Helix spacing |
| $\Delta B$ | Minimum element size |
| $D_r$ | Density |
| $\varphi_0$ | Initial internal friction angle |
| $\varphi_{cr}$ | Critical internal friction angle |
| $\varphi_p$ | Peak internal friction angle |
| $\psi_p$ | Peak dilatancy angle |
| $\sigma_3$ | Confining pressure |
| $\varepsilon_d^r$ | Critical equivalent plastic strain |
| $\varepsilon_d^p$ | Peak equivalent plastic strain |
| $E$ | Young's modulus |
| $v$ | Poisson's ratio |
| $N_\gamma$ | Uplift capacity factors |
| $\gamma'$ | Soil effective unit weight |
| $A$ | Plate area |
| $Q_u$ | Ultimate uplift capacity |
| $u_f$ | Failure displacement |
| $z$ | Vertical distance from the lowermost plate |
| $z_p$ | Vertical distance of the peak value point from the plate |
| $\alpha$ | Inclination between the simplified rupture plane and the vertical direction |
| $\sigma_n$ | Normal stress |
| $K_0$ | Initial lateral pressure coefficient |
| $K_u$ | Lateral earth pressure coefficient |
| $K_n$ | Lateral earth pressure coefficient proposed by Hao [12] |
| $K_{u,peak}$ | Peak lateral earth pressure coefficient |
| $\varphi^*$ | Internal friction angle of the equivalent associated plastic MC model |
| $\varphi^*_A$ | Average internal friction angle |
| $W$ | Soil weight |
| $l_i$ | Vertical height of the $i$th rupture surface |
| $r_{ib}, r_{it}$ | Bottom and top radii, respectively |

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
