# Peer review of "Numerical Investigation of Uplift Failure Mode and Capacity Estimation for Deep Helical Anchors in Sand"

_jmse, doi:10.3390/jmse11081547_

Round 1
Reviewer 1 Report
All figures are in very low quality; the size is too small.
Line 114: use "density" instead "compactness".
Lines 151 and 157: Please, make clear that the referred plastic strain in, in fact, shear (deviatoric) strain.
Line 205: does it hold for non-log axis? avoid plotting these curves in logarithm scale. It is not usual and it masks tradicional features takens from these curves, as initial modulus, peak load etc.. in the axis legend use parethesis for the unity not slash bar.
Line 239: is this plastic strain? or strain during plastic flow? in hardening materials they are not the same.
Line 410: Please, clarify if associated or non-associated flow-rule was considered and make some comment about the consequences in considering one of them.
Figure 14 and Figure 15: Use the same scale in order to make direct comparison easier.
Line 494: use only one decimal algarism.
General comments:
- all figures are with very low quality. Increase the size.
- Avoid using: Loose dense sand - Use "loose sand" only. Also avoid using extremely dense sand and mediu dense sand. Use dense and medium. It is enough!
Reviewer 2 Report
The paper approaches an interesting topic which is not so much approached.
Paragraph 2.1 and after - it is recommended to use "-" instead of ~
for value range.
The numerical results are compared to those obtained by other authors from centrifuge testing, but it is not clear if the sand used in the numerical model is the same as the one in centrifuge testing. The paper refereced as [12] is mentioning only general properties of the used sand and the critical friction angle. In addition, the centrifuge test was conducted only on dense sand.
It is affirmed in the paper that "Young's modulus E and peak internal friction angle φp is taken from the literature [48], which are obtained from triaxial tests for the sand." But, in [48] there is a sand with different properties than for [12] and I didn't find values for E and fi..
Values of E and fi for other relative densities was determined also from litterature using a linear interpolation.
But, when performing centrifuge tests all necessary geotechnical parameters are determined using triaxial tests, it is a special sand which is used, therefore I cannot understand why the parameters were determined from litterature and were not taken the exact values from the centrifuge tests..
There also other parameters attributed to [48], but I couldn't find them in that paper.
More details and precisions should be given about the values of the geotechnical parameters. Also, their evolution with the depth is quite strange for fi and psi.
Why gamma' was used in the calculation of the uplift capacity? There is groundwater considered?
The comparison with the centrifuge test results is very much affected by the problems in chosing the parameters.
Section 4,3 is quite unclearly presented, using many new symbols and parameters. A better systematisation of this sections would be useful to optimize the reader's understanding.
English expression is fine, only minor errors detected
Reviewer 3 Report
The paper contains an interesting material and presents analytical and numerical modeling of deep helical anchors in sand. To verify the results, the authors make a comparison with the results obtained by experimental modeling in the centrifuge. Based on the numerical modeling performed, the authors make a proposal for a method of calculating the bearing capacity of anchors in sand based on limit equilibrium.
The positive points of the paper:
- a good overview of the calculation methods of helical anchors and failure mechanisms;
- logical development of research starting from bibliographic data, then numerical modeling in finite element performed respecting the knowledge accumulated based on bibliographic research; comparison with results obtained by experimental modeling in centrifuge and finally proposal of a relatively simple method for estimating the tensile bearing capacity of helical anchors.
The negative points:
- the fact that the geotechnical parameters of sand are estimated based on literature data. It is not understood why the parameters of the soil did not use the centrifugal tests with which the results were compared. Certainly, the sand used for centrifuge tests is studied and all mechanical parameters are known. Under these circumstances, the comparison between numerical modeling and centrifuge modeling can no longer be very reliable.
- the failure mechanism of the anchors is different for shallow anchors and for deep anchors. But it is not indicated in the paper what are the criteria that classify anchors in shallow or deep anchors.
Table 1.
Geotechnical parameters of sand estimated based on literature data and empirical relationships are presented in Table 1. However, the inverse variation of the values of the friction internal angle and the dilatancy angle with depth is not clear. The deformation modulus increases with depth, which is correct.
Line 437
The exponential decrease rate (k) is missing.
Line 449
The title of chapter 4.4 is wrong (identical with chapter 4.3)
Lines 476-477
References should be made to equation 11, not 10.
Round 2
Reviewer 2 Report
The paper has been a little revised in terms of text, the citation of references in the text is improved. Also, the list of notations is useful .
But, the main issues were not addressed in the paper. In their response to the reviewer there are a lot of explanations, especially regarding the sand parameters related to the cited papers, but no improvement in the paper. It mhas been added only that the sand is the same as in one or more cited papers, but no evidence was found. From the text I have the impression that "medfium sand"means medium dense sand, while normally medium sand meand medium from a grain size distribution point of view. The text is very vague ("relatively close values"). Therefore, I couln'd find a real improvement of the paper with respect to the explanation of parameters, their complete presentation etc.
Section 4.3 benefits of the list of symbols, which improves the readability, but there is no significant change in the text for a better organisation and a more understandable text.
For gamma' I received an explanation, but there is no value for gamma or gamma' in the text, for seeing that the proper value has been used.
English is fine, only minor editing required
Reviewer 3 Report
The authors responded to the comments.
Author Response
Thank you for your advice.
Round 3
Reviewer 2 Report
The paper was slightly improved in terms of how the approach is presented. It is clearer now how the authors considered the geotechnical parameters. However, it is also clear that there were no triaxial tests for determining the parameters for other relative density values and there are no sufficient proofs for the empirical formulae used for determing them. The main weakness of the paper is this one, meaning that the comparison is made using geotechnical parameters determined in various manner. And this is fundamental and there is no significant change, as it cannot be done right away and because there is a lack of reliability in the values! If this can be done and together with the revision that was already done, the paper will become a very good and publishable one, especially if the organisation of the paper will be also improved.
The term "medium sand" has to be replaced by "medium dense sand" as this is misleading.
There is still no explanation for the strange evolution of fi and psi with the depth.
English expression is fine
